# Bird Satellite Tracking Revealed Critical Protection Gaps in East Asian–Australasian Flyway

**DOI:** 10.3390/ijerph16071147

**Published:** 2019-03-30

**Authors:** Jialin Lei, Yifei Jia, Aojie Zuo, Qing Zeng, Linlu Shi, Yan Zhou, Hong Zhang, Cai Lu, Guangchun Lei, Li Wen

**Affiliations:** 1School of Nature Conservation, Beijing Forestry University, Beijing 100083, China; leijialinbjfu@foxmail.com (J.L.); jiayifei@bjfu.edu.cn (Y.J.); holidayzuo@126.com (A.Z.); zengqing@bjfu.edu.cn (Q.Z.); linlu.shi@foxmail.com (L.S.); lucai.wetland@foxmail.com (C.L.); 2Co-Innovation Center for Sustainable Forestry in Southern China, College of Biology and the Environment, Nanjing Forestry University, Jiangsu 210037, China; zhouyan.eco@foxmail.com; 3East Dongting Lake National Nature Reserve Authority, Yueyang, Hunan 414000, China; zhanghong258@sina.com; 4Science Division, Office of Environment and Heritage, Sydney, NSW 2141, Australia

**Keywords:** migration route, stopover, utilization distribution, Croplands, Northeast China Plains, Bohai Bay

## Abstract

Most migratory birds depend on stopover sites, which are essential for refueling during migration and affect their population dynamics. In the East Asian–Australasian Flyway (EAAF), however, the stopover ecology of migratory waterfowl is severely under-studied. The knowledge gaps regarding the timing, intensity and duration of stopover site usages prevent the development of effective and full annual cycle conservation strategies for migratory waterfowl in EAAF. In this study, we obtained a total of 33,493 relocations and visualized 33 completed spring migratory paths of five geese species using satellite tracking devices. We delineated 2,192,823 ha as the key stopover sites along the migration routes and found that croplands were the largest land use type within the stopover sites, followed by wetlands and natural grasslands (62.94%, 17.86% and 15.48% respectively). We further identified the conservation gaps by overlapping the stopover sites with the World Database on Protected Areas (PA). The results showed that only 15.63% (or 342,757 ha) of the stopover sites are covered by the current PA network. Our findings fulfil some key knowledge gaps for the conservation of the migratory waterbirds along the EAAF, thus enabling an integrative conservation strategy for migratory water birds in the flyway.

## 1. Introduction

The conservation of migratory birds has gained new momentum across the globe [1], mainly due to the recognition of their irreplaceable ecological functions and services [2], but also because of the widespread declines in their populations [3], and more sophisticated tools to track their movement across the landscape [4]. The scale and intensity of threats on migratory birds have increased since the 1980s with rapid economic development and human population growth [5,6]. These threats include climate change [7,8], habitat loss, degradation and fragmentation [9], over-harvesting and illegal hunting [10], pollution and invasive species as well as emerging diseases [11], among others.

Like other migratory animals, migratory birds travel long distances in order to avoid competition, escape predation, breed, and to take advantage of spatiotemporal resource variation. However, the mortality of migratory birds during their journey poses a unique challenge for conservation biologists, which requires a multitude of approaches to address the threats encountered along their migration routes [1]. Conservation actions targeting part of life circle (e.g., breeding, nonbreeding or stopover sites) may not be able to prevent population decline of these birds [12,13].

Almost all long-distance migrants use of stopover sites [14], even for bar-tailed godwits (*Limosa lapponica*) (Linnaeus, 1758), the longest nonstop migratory bird on records [15]. Stopover sites are habitat patches used by animals as refueling stations and temporary rest sites along the migration journey between breeding and nonbreeding ranges [16,17]. The availability and quality of stopover sites have a large influence on the rate of mass gain [18], which can in turn have implications not only for the mortality of the migratory birds during migration [5], but also for the reproductive success in the coming breeding season [19,20], and survival during the nonbreeding season [21]. Indeed, stopover ecology has become an intense research area in avian ecology, especially for songbirds [22,23]. and shorebirds [24]. However, information regarding the stopover site availability, extent, and habitat quality is still largely overlooked in developing conservation strategies for migratory animals [14] apart from few cases of migratory shorebirds [25]. This is particularly true for the East Asian–Australasian flyway (EAAF). As one of the EAAF current partners, China has done lots of work to support the conservation objectives. Although there are focused studies on the main wintering sites such as Poyang and Dongting Lake in the middle and lower Yangtze regions [26,27], large-scale and long-term bird population surveys, as well as breeding and nonbreeding (including stopover sites) habitat assessments are largely missing [13].

Understanding migratory connectivity, i.e., the geographic linking of individuals and populations throughout the annual circle [28], is fundamental for the conversation of migratory birds. However, current conservation strategies for migratory species often fail to take into account migratory connectivity, thus limiting the efficiency of current conservation efforts [14]. In comparison with breeding and wintering distribution, stopover ecology of migratory Anatidae in East Asia is severely under-studied. In general, we do not know the timing, intensity and duration of stopover site usages by many migratory Anatidae. Data and information that reflect the migration patterns, location, extent and habitat quality of key water bird stopover sites are not available [29]. The lack of detail knowledge on stopover ecology prevents the development of effective and full annual cycle conservation strategies for migratory Anatidae [30] and this represents the weakest link in integrative conservation network in EAAF. 

Rapid advances in technologies, such as satellite transmitters, global positioning system (GPS) loggers, light level loggers [31], DNA markers [32], and stable isotopes [33], provide an excellent opportunity to study migratory birds throughout their annual cycle [34]. Concurrent with the advances in spatial tracking technology (e.g., GPS), many quantitative methods have been developed to quantify temporal space use through calculating utilization distribution (UD). These methods can be broadly divided into two groups: the classic location-based kernel density estimation (LKDE) [35], and the movement-based kernel density estimation (MKDE). The MKDE is superior to LKDE for higher temporal resolution telemetry data as the approach can incorporate time, distance, measurement error, uncertainty of the movement between two consecutive locations, and habitat into estimating UD [36].

The East Asia populations of waterfowl Anatidae, including Lesser White-fronted Goose (*Anser erythropus*, Linnaeus, 1758. Referred to as LWFG thereafter), Greater White-fronted Goose (*Anser albifrons*, Scopoli, 1769. GWFG), Bean Goose (*Anser fabalis*, Latham, 1787. BG), Greylag Goose (*Anser anser*, Linnaeus, 1758. GG) and Swan Goose (*Anser cygnoides*, Linnaeus, 1758. SG), have experienced severe declines in the past 50 years [37]. These migratory herbivores breed in temperate zones (including Northern China, Mogolia and Southern and Central Siberia, mainly GG, BG and SG) or subarctic zones (mainly LWFG and GWFG) [38,39]. During the non-breeding season, these birds winter in large lakes and wetland complex in East Asia, especially the mid-lower Yangtze regions [40]. We expect similar threats that have been identified in other flyways such as the European–African and America migration systems to be prevalent in EAAF [1], however, we are unclear in relation to the main drivers of their population decline. We virtually have no data to refer the relative contribution of reproductive and survival at breeding range, mortality at wintering grounds and stopover sites to the flyway population dynamics.

Wetlands in Eastern China forms part of the core areas of the EAAF. However, the rapid change of land cover, especially the expansion of urbanization, may pose a grave threat to the migratory species [5]. For example, as one of the main agricultural regions in China, 30% of the land in the Northeast is arable, providing abundant food resource for migratory waterbirds [41]. However, the Northeast has experienced land cover change from 1990 to 2000, with farmland decreased by 386,195 ha and grassland decreased by 140,075 ha [42]. With the loss of natural wetland and human induced land cover conversion, the protection of stopovers in Eastern China is facing a great challenge.

In this study, we focused on the spring migration as spring stopovers are critical for the population dynamics of Arctic geese [43], as they affect directly the reproduction success and survival of juveniles. We estimated the relative amount of land use along the migration route, which thus provides the foundation for conserving migratory routes and allows the quantitative assessment of current conservation efforts. Our hypothesis is that the spring stopovers of migration geese are mostly unprotected giving the rapid land cover alternation in the last 30 years [42], and the objectives of the study include: (1) determine the key stopover sites of five geese species in China; (2) the analysis of the main landscape type of those key stopover sites; and (3) evaluate protection gap and identify the critical shortfalls in Eastern Asian water bird conservation network. Furthermore, we estimated the relative amount of land use along the migration route through combining multiple individual paths into a population-level estimate of migration corridors. Delineation of the population-level migration route provides the foundation for conserving migratory routes and allows the quantitative assessment of current conservation efforts. The objectives of the study include: (1) determine the key stopover sites of five geese species in China; (2) the analysis of the main landscape type of those key stopover sites; and (3) evaluate protection gap and identify the critical shortfalls in Eastern Asian water bird conservation network.

## 2. Materials and Methods

### 2.1. Ethics Statement

We declare that all field methods used in this study were approved by the Forestry Department of Hunan Province under scientific research license (No.11 Xiang Forest Protection (2014)). Field research was conducted with permission from the Bureau of East Dongting National Nature Reserve.

### 2.2. GPS Tracking

This is part of a larger study on the waterfowl migration network in EAAF. In the winters of 2015, 2016 and 2017, a total of 141 geese from five species, including LWFG, GWFG, BG, GG and SG, were captured and tagged, unharmed by experienced hunters using baited clap traps in East Dongting Lake and Poyang lake (Appendix A
Figure A1). These two lakes are the most important wintering sites for migratory geese in EAAF, supporting more than 70% of Eastern Lesser White Fronted Goose population [44,45].

Healthy adult geese were tagged with GPS transmitters (Hunan Global Messenger Technology Company, China). The solar-powered global system for mobile communication (GSM) transmitter has an integrated general packet radio service (GPRS) subsystem for short message service (SMS). Due to financial restriction, international data-roaming function was not enabled so that no real-time data was sent once the bird was outside of China. The data collected outside of China were downloaded once the bird returned to China in the following autumn migration. The satellite transmitter was small (55 × 36 × 26 mm) and light (22 g, between 0.6 and 1.6% of the bird’s body weight), therefore well below 3% of body weight [46], and had neglectable long-term habituation effect on goose behavior [47]. We followed every tagged bird closely for the entire period when they were in China, any abnormal situations such as loss of transmission (death or device failure), long-time stationary (death or injury), were investigated on ground. The sampling intervals were programmed to be 1–3 h. However, the intervals could be changed manually via an online panel according to battery usage, solar conditions and locations to optimize data delivery.

There are five class of the spatial accuracy (A, B, C, D and null correspond to 1-sigma error radius of <10 m, 10–100 m, 100–1000 m, and >1000 m, and not available, respectively). We included classes A, B and C in this study. We included class C (i.e., accuracy 100–1000 m) as the mean distance between a pair of consecutive fixes was much greater than 1 km (Appendix A
Table A1).

### 2.3. Delineating Migration Routes and Stopover Sites

As temporal autocorrelation is common, or even an intrinsic property of animal relocation sequential data [48], we used the biased random bridge (BRB) approach to estimate the UD of individual goose at the same spatial extent and at a 4 km^2^ grid resolution. This approach is similar to the Brownian bridge approach [49], with several improvements [50]. A Brownian bridge estimates the density of probability that a trajectory passes through any point of the study area. The Brownian bridge is built on a conditional random walk between successive pairs of relocations, dependent on the interval between locations, the distance between locations, and the Brownian motion variance that is related to the animal’s mobility. The Brownian bridge approach supposes that the animal movement between two consecutive locations is random and purely diffusive: it is presumed that the animal moves in a purely random fashion from the starting relocation and reaches the next relocation. The BRB approach goes further by adding an advection component (i.e., a “drift”) to the purely diffusive movement in the Brownian bridge; it is presumed that the animal movement is governed by a drift component (i.e., a general tendency to move in the direction of the next relocation) and a diffusion component (tendency to move in other directions than the direction of the drift). The addition of the drift allows the BRB approach to model animal movements in a more realistic manner [51].

Considering one step in an animal trajectory including two successive relocations r_1_ = (x_1_, y_1_) and r_2_ = (x_2_, y_2_) collected at time t_1_ and t_2_, the BRB estimates the probability density function (PDF) of the animal located at any place r_i_ = (x_i_, y_i_) at time t_i_ (with t_1_ < t_i_ < t_2_). Benhamou [36] noted in 2011 that the BRB can be approximated by a bivariate normal distribution:(1)$$\begin{array}{c} f\left(r_{i},t_{i}|r_{1},r_{2}\right)=\frac{t_{2}-t_{1}}{4\pi D\left(t_{2}-t_{1}\right)}exp\left[\frac{r_{m}Dr_{m}}{4p_{i}\left(t_{2}-t_{1}\right)}\right]\\ r_{m}=\frac{x_{1}+p_{i}\left(x_{2}-x_{1}\right)}{y_{1}+p_{i}\left(y_{2}-y_{1}\right)}\\ p_{i}=\frac{t_{i}-t_{1}}{t_{2}-t_{1}}, \end{array}$$
where r_m_ is the mean location, p_i_ is the proportion of time from starting relocation to r_m_, and D is the diffusion matrix. A more detailed description of BRB approach can be found in [36].

We used the complete sequence of relocations that occurred between the wintering and breeding grounds to estimate UDs for each spring migration route (a total of 33, Appendix A
Table A1). For each bird, we defined UD values within the 50% contour were stopover sites used for resting and feeding, and those between the 50% and 95% contours were flight paths with minimal stops [52]. However, this process produced many stopover sites along the migration paths and most of the sites were very small (e.g., 108 stopover sites were identified for the longest recorded migrant LWFG12.17 and the median size was 0.03 ha). In order to reduce the number of identified stopover sites, we applied a three-day rule to validate the identification, i.e., the bird should have spent at least three days within a site to be classified as a stopover. We acknowledge that the three-day rule was rather arbitrary, nonetheless, this rule produced promising results. For example, by applying the three-day filter, there were eight stopover sites (the minimum site was 657.25 ha) along the migration route of LWFG12.17, and collectively the sites covered 88.9% of the recorded fixes. In contrast, when applying a five-day filter, there were only two stopover sites identified with average size of 2553 ha and counted for only 51% of the GPS locations.

We summed the 33 BRB models to delineate the population level stopover sites and migration route. The summed UD was then standardized so that the pixel value was in the range of 0–100. We applied Jenks natural breaks method [53] to classify the UD into stopover sites and migration path. The resultant raster was transferred into a polygon shape file and used for habitat use and protection gap analysis. All spatial data were projected using the world azimuthal equidistant coordinate system to calculate areas and distances.

### 2.4. Habitat Use at Stopover Sites and Protection Gap Analysis

We obtained the 2015 land cover (LC) map for Eastern Asia from European Space Agency Climate Change Initiative (ESA CCI), which delivers annual global LC maps at 300 m spatial resolution from 1992 to 2015. We aggregated the LC classes within our study region in the ESA CCI system into six broad land use types: cropland (A), wetland (including estuary) (B), grassland (C), forest (D), bare-ground (E), and build-up area (F). We summarized the LC for the three migration passages delineated in the study. We also calculated the percentage of landcover classes within each delineated stopover site by overlapping the stopover polygons on the LC raster to investigate the habitat preference.

We compared the identified stopover sites with the World Database on Protected Areas (WDPA) to investigate the migration protection gap. The WDPA is the most comprehensive global database of marine and terrestrial protected areas and is one of the key global biodiversity data sets being widely used to inform planning, policy decisions and management. It includes a wide range of protected areas that meet the IUCN and Convention on Biological Diversity (CBD), therefore, it can be used to identify the protection gap in stopover sites. We calculated the protection gaps by overlaying the identified stopover sites with WDPA Dataset in ArcGIS 3.2 (ESRI, Redlands, CA, USA). We applied the following criteria to classify the stopover sites as below:>75%    Totally/satisfactorily protected;25–75%  Under-protected;<25%  Critically under-protected; and=0     unprotected.

## 3. Results

### 3.1. General Description of Spring Migration of Eastern Geese

Many of the tracked birds (over 80%) were lost mainly due to device failure and death. As the GPS sends real-time data only within China, we could not estimate the global survival rate of the geese, as well as the overall percentage of loss due to death or device failure from this study. However, with the 80 cases of loss within China, we estimated the percentage of loss due to device failure to be as high as 81% (i.e., 46% of device failure rate). Out of the 15 cases of confirmed death, one third were diagnosed as poisoned or hunting, and we did not find any death was directly caused by tagging (Appendix A
Table A2).

Out of the tagged geese, we obtained 33 completed spring migratory paths (four for BG, five for GG, 10 for GWFG, 12 for LWFG and two for SG. Note that some birds had multiyear routes, see Appendix A
Table A1). The movement patterns (e.g., the timing of departure from the breeding grounds and arrival at the wintering sites) suggested that none of these tagged birds were from the same family. Therefore, these trajectories were representatives of the flyway. The geese displayed variation in migration behavior in terms of timing of onset, direction, distance, duration and speed between individuals and species, and between different years of migration made by the same goose. In general, LWFG, GWFG and BG travelled longer and faster than Greylag and SG (Appendix A
Table A1). The onset of migration was generally in late March but could be as early as 11 February 2017 (Greylag02) and as late as 24 April 2017 (SG05). The dates arriving at the breeding grounds varied from the 1 April 2016 (Greylag04) to the 3 July 2017 (SG05). The longest migration lasted for 92 days from 4 March 2016 to 4 June 2016 (LWFG12) while the shortest duration lasted for only 33 days (SG04, from 8 March 2017 to 10 April 2017). The longest migration was made by a Less White Fronted Goose (a total of 16,172 km in 60 days), and the shortest one was made by a Greylag Goose (2458 km in 34 days).

### 3.2. Migration Paths and Stopover Sites

All the tracked birds started their migration flying northeasterly to the coastal wetlands around the Bohai Bay, such as the Yellow River Delta, Shuangtaizi Estuary, and Yalujiang Estuary (Figure 1a) before dividing into separated routes to their breeding sites. From Bohai Bay estuaries, two migration routes were notable based on the geography (Figure 1b): the LWFG, GWFG, and some of BG took the northeast passage (eastern of the Greater and Lesser Khingan Range) to breed at the Arctic tundra as Kytalyk, Keremesit-Sundrun Catchment, Lena Delta and Lower Anadyr lowlands; and the SG, Greylag and some individual of BG took the northwestern paths to their steppe breeding grounds in the Mongolian-Manchurian grasslands such as Mongol Daguur and the Ulgai Lake (Appendix A
Figure A1). The northeastern passage is generally flat and has low elevation (less than 500 mASL) while the northwest route passes though mountain ranges and has a high elevation (greater than 1000 mASL) (Figure 1b). Based on the temporal and geographical information, we distinguished three migration passages: the southern (i.e., from wintering sites to Bohai Bay), the northwest and the northeast divided by the Greater and Lesser Khingan Ranges.

For individual routes, we identified a total of 112 stopover sites with a total area of 2,192,823 ha (Appendix A
Table A3). The geese had distinct stopover ecology in terms of the number of stopover sites (ranging from one to eight), extent of the sites (mean ± sd = 387,748 ± 308,684 ha), and usage of stopover sites (total days spending in stopover site ranged from 13 to 82) (Appendix A
Table A3). Also, the average days they spent and the average range of motion on single stopover are quite different (Figure 2). On average, these geese spent 84% of their migration time in these delineated stopover sites (ranged from 53–97%), while the rest was for flight and short-duration stay. Most of the stopovers were within China (110 out of 112), where the birds took a more or less ‘direct’ flight onwards to their breeding sites at the temperate steppes (Greylag and SG) and arctic tundra (BG, GWFG and LWFG) without extensive stopovers (in terms of both duration and spatial extent). As human intervene is low in the Arctic tundra, the following protection gap analysis was focused on China.

These delineated stopover sites varied greatly in shape and extent (ranged from 103–269,159 ha, Figure 1b). Generally, the stopovers in the northwestern routes were much smaller than those at the southern and northeastern passages (mean size is 14,445, 48,483 and 4476 ha for southern, northeastern and northwest stopovers, respectively, Table 1). In addition, the habitat utilization was much higher in the northeast routes than in other routes (Table 1). The tracking records also indicated that the GWFG and LWFG never used the northwest stopovers and Greylag and SG never used the northeast stopovers. Most of the earlier southern stopover sites were located around Bohai Bay, especially the Yellow River Delta, whereas all the northeast stopover sites were within Northeast China Plains, including Sanjiang Plain, Songnen, Songliao Plain, and Liaohe Plain (Figure 1b).

### 3.3. Habitat Use and Gaps in Protection

Overall, croplands are the largest land use type in the stopovers followed by wetlands, and natural grassland (Table 2). The three land use types accounted for 96.27% (or 2,111,031 ha) of the total size of the stopover sites. There were dramatic differences in habitat uses between the three passages, especially for the stopover sites along the northwest routes (Table 2). In the northwest stopover sites, the largest land use type is grasslands followed by croplands and wetlands.

Only 20 of the 110 identified stopover sites in China overlap with the WDPA, leaving the majority (nearly 82%) unprotected (Table 3). In terms of area, nearly 85% of the stopovers were not protected, only 0.14% had satisfactory protection (Table 3). Spatially, stopovers within the northeastern passage had the highest proportion intersected with WDPA. For the Southern passage, 28.18% of the stopover site area was covered by WDPA, and the majority (58.48%) were the estuaries around the Bohai Bay (Figure 1b), notably the Yellow River Delta, Liaohe River Estuary and Inner Gulf of Liaodong. Within the Northwestern passage, less than 3% of stopover site area was protected by WDPA.

## 4. Discussion

In the study, we focused on the spring migration as we did not receive enough autumn migration data due to mortality and device failure (Appendix A
Table A2). In addition, spring migration has direct effects on reproduction, and is thus more important for the fitness of the species [41].

The tagged geese displayed various migration behaviors between species, among individuals from the same species, and between different years of migration made by the same goose. The behaviors differed in onset, distance, duration, flight direction and speed. Our results demonstrated that the geese wintering in the Yangtze floodplains shared a common path initially to the wetlands and estuaries around Bohai Bay, from where two geographically distinct passages could be identified: (1) the narrower northwest passage was mainly taken by the short-distance migrants (i.e., Greylay and Swan Goose, average travel distance less than 4500 km) to their breeding grounds in the temperate steppes; and (2) the more dispersed northeast was taken by long-distance migrants such as Lesser White Fronted Goose (mean distance 13,453 km) and Greater White Fronted Goose (mean distance 8658 km) to their breeding range in Arctic and Sub-Arctic tundra. However, the Bean Geese, which has the largest breeding distribution among the five species [54], could use both passages to reach their breeding ground in taiga or tundra (mean migration length 6655 km).

Geese are early migrants [55]. Most of the tagged geese departed the wintering ground at late March (i.e., early spring in the Yangtze region), which is synchronic to the water level raising in their major wintering habitats (e.g., Dongting and Poyang Lake) [40,56]. Nevertheless, the onsets of spring migration varied considerably from 11 February 2017 (Greylay Goose) to 24 April 2017 (Swan Goose), and the migration duration also differed substantially ranging from 33 (Swan goose) to 92 days (Lesser White Fronted Goose).

More than half of the core stopover sites were croplands. In the more intensively used northeastern passage (Table 1), the percentage of croplands was even higher (72.00%). These findings indicated that the geese tend to feed on croplands for refueling during their long journey. Studies in Europe [57] and North America [58] have linked the geese population increase since the 1960s to the expansion of agricultural land usage [59]. Crops, such as wheat, maize, oat and barley, provide superior energy and nutrient content than natural foods [60]. Moreover, the intake rates are higher for crop offering more rapid accumulation of daily needs [59]. Although our results display strong evidence that geese intensively used the croplands as their stopover sites to supplement their body reserve, we cannot establish the causality between abundance of high quality food and habitat preference of geese. The congregation of geese on croplands might simply be due to the rapid loss of natural habitats to agriculture [61]. For example, Sanjiang Plain, where a large number of stopover sites were identified (Figure 1), was historically covered by lashing marshes dominated by *Carex lasiocarpa* community [62]. However, since the 1950s, most of wetland (70%) and natural grasslands (98%) have been converted to croplands for grain production [63]. The higher percentage of cropland usages might just reflect the landscape composition of landcover. In addition, the grassland and wetland usage (23.88%, Table 2) represented more than 65% of the total wetland and natural grassland in the region (in comparison, this value was only 5% for croplands). This suggests that the geese have attempted to explore most of the natural habitats, and might indirectly imply that the geese preferred the natural environment as opposed to the croplands. Detailed movement study with fine temporal resolution telemetry data could provide insights into geese habitat preferences.

The timing of stopover was predominantly from earl April to early May, which is overlapped with the onset-of-growth for most crops in Northeast China Plains [64]. Within this time window, intensive human activities (such as ploughing, sowing, fertilizing, and weeding) forced birds to spend more time in vigilance and escape flight at the cost of foraging activities [65]. The human disturbance negatively impacted the energy reserve and body condition, and subsequent breeding success of the geese [43]. In the field survey, we found dead bodies of LWFG, GWFG, and Mallard (*Anas platyrhynchos*) (Linnaeus, 1758) with no obvious injuries in croplands in Northeast China Plains. An interview with local residents found that bird deaths during stopovers occurred every year although no large-scale mortality in the region was reported. The geese had rather low natural mortality rates once they fledged this area [66] implying that the likely causes of the death could be agricultural chemical residuals, human disturbances and poisons intentioned by illegal hunters. Geese in the agricultural landscape are more likely to face threats because of the intentional and unintentional anthropogenic impacts on the environment. The loss of natural wetlands and threats in cultivated land could have dramatically reduced the availability of suitable habitat for migration geese and adversely affected their survival rate. Hence, a detailed management proposal for main stopover sites of geese, especially integrating wildlife conservation into agricultural practices, is imperative to safeguard the population stability. This is an overlooked issue in current conservation policy in China, and should be a key element of migratory connectivity protection.

Our results showed that there were considerable spatial variations on stopover protection within the three migration passages. A very small proportion (about 5%) of stopover sites identified in this study in the northwestern passage is currently covered by PAs. Although reclamation and agriculture expansion are limited and (semi-)natural grassland is the main landcover type in this region, land degradation due to over-grazing [67] and development of wind power infrastructure [68] are among the major threats for migratory bird. Stopovers in the southern passage (i.e., from the wintering grounds in the lower middle Yangtze region to the coastal estuaries around Bohai Bay) were relatively well protected compared with the northeast and northwest passages. Within this passage, many of the stopover sites (notably the high-profile Yellow River Delta), are key sites for shorebird conservation [69]. Therefore, management actions, such as maintaining and restoring tidal flats, are conservation priority. Our study demonstrated that these sites were also intensively used by migratory geese, especially by Swan Geese. Thus, an integrated management strategy, which incorporates the food requirements of the migratory Anatidae, such as sustaining the coastal brackish wetland, is urgently needed. In the most intensively used stopover sites in Northeast Plains, less than 20% (243,024 ha) were located in protected areas (PA). Note that several PAs in this region, such as Momoge Nature Reserve, Zhalong Nature Reserve and Xianghai Lake, are also key stopover sites for oriental white storks (*Ciconia boyciana*, Swinhoe, 1873), Siberian cranes (*leucogeranus leucogeranus*, Pallas, 1773), and hooded crane (*Grus monacha*, Temminck, 1835) [68]. To enhance the conservation of migration connectivity, it might be necessary to expand these key PAs and combine adjacent nature reserves.

Many studies have demonstrated that stopover sites are critical for the population dynamics of migratory geese due to the great influence on breeding success [19,70]. However, despite the EAAF having the highest proportion of threatened migratory birds (19%) [71], most researches on stopover ecology originate from northwest Europe and North America, and very few have been carried out in Asia [59,72]. The knowledge gap of the migration connectivity of the Eastern Anatidae, which breed in temperate steppe and arctic tundra and winter in the (sub)tropical Asia [54], might impede the global efforts on conservation of migratory birds. This study utilized the multiyear (2015, 2016 and 2017) and multi-taxa (five species) telemetry data to identify and delineate the spring geese migration paths and stopovers in Eastern Asia. Furthermore, we assessed the gaps in the current conservation network through overlaying the delineated stopover sites and World Database on Protected Area (PA), which represents the current protection status.

The designation and effective management of protected areas are global priority for preventing and slowing biodiversity loss [73]. WDPA is the most comprehensive global database of marine and terrestrial protected areas, and is one of the key global biodiversity datasets being widely used. However, our analysis demonstrated that the migration connectivity of Eastern geese was severely under-protected by the current protected areas. Of the 110 identified stopover sites in China, only 20 (or 15.63% in area) intersected with PAs and the majority was outside of PA boundaries. In addition, the size of most PA is smaller than the daily foraging range of goose. In terms of protection levels, only 0.14% of the stopovers had satisfactory coverage under the current PA network. As a result, the geese were more likely be threatened by range of risks such as food resource shortage, habitat degradation, and poison and poaching.

## 5. Conclusions

In the study, we delineated three distinct migration passages: all the tagged geese shared the common southern passage to the wetlands and estuaries around Bohai Bay, from where short-distance migrants such as Swan Goose and Greylag Goose took the northwest routes to the temperate steppes; and the long-distance migrants (i.e., Lesser White Fronted Goose and Greater White Fronted Goose) flied northeast to the arctic tundra. Most of the stopover sites were in China, from where the geese travelled more or less directly to their breeding destinations. Stopovers sites were mainly covered by croplands and were severely under-protected by the current protection network. The revealed protection gaps may have contributed to the high proportion of threatened Anatidae species in the EAAF. In addition, our study could not establish the causality between agricultural land usage and goose habitat preference despite the tagged geese heavily used croplands.

Expanding the current protection network and integrating wildlife conservation into agricultural practices are critical for conserving the migratory waterfowl. Our findings fulfill some key knowledge gaps for the conservation of the migratory water birds along the EAAF, and allow the quantitative measure of migratory connectivity, and thus enable an integrative conservation strategy for migratory waterbirds in the flyway.

## Figures and Tables

**Figure 1 ijerph-16-01147-f001:**
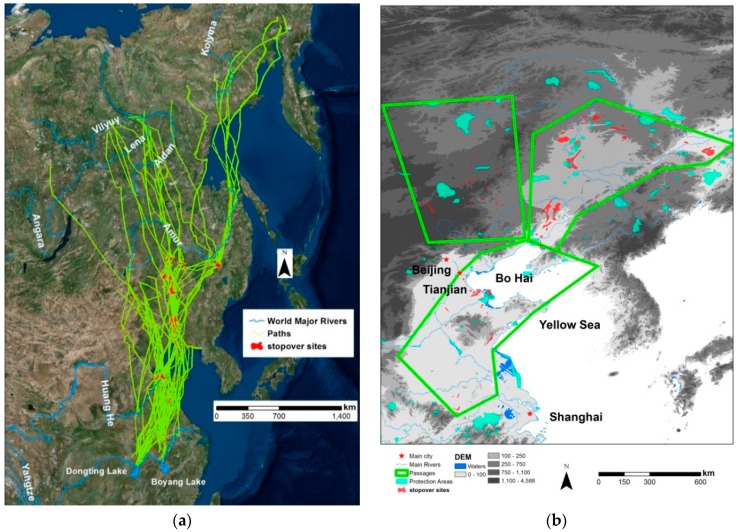
Spring migration paths and stopover sites of the 33 tagged geese (**a**). The geese were caught and tagged at their wintering ground of Dongting and Poyang Lake. The 33 paths are the 50–95% contour of utilization distribution (UD); and stopover sites are the the 50% contour of summed UDs based on Jenks natural breaks. The three migration passages, the location of China’s stopover sites and their spatial relationship with protection areas (**b**). Maps were produced using ArcGIS (v 10.3, Esri, Redlands, CA, USA) with world image (source: Esri, DigitalGlobe, GeoEye, Earthstar Geographics, CNES/Airbus DS, USGS, USDA, AeroGRID, IGN, and the GIS User Community).

**Figure 2 ijerph-16-01147-f002:**
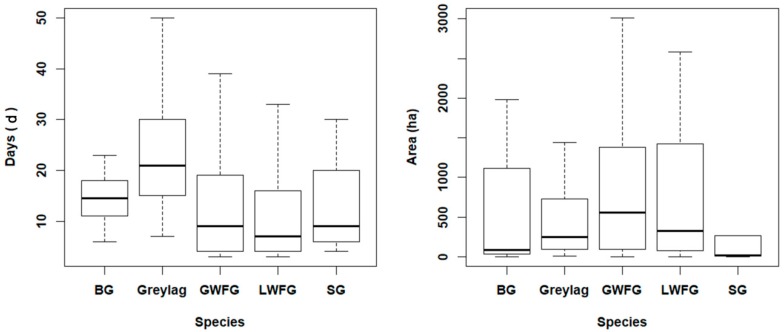
Summary graphs of the average days the geese spent in stopover site (**left**) and the home range the geese used in stopover site (**right**). The box-whiskers graphs show the median (bar), upper and lower 25% percentiles (box) and range (whiskers) of the days (**left**) and area (**right**) of the average single stopovers for Bean Goose (BG), Lesser White-Fronted Goose (LWFG), Greater White-Fronted Goose (GWFG), Greylag Goose and Swan Goose (SG).

**Table 1 ijerph-16-01147-t001:** Summary of geese spring stopover sites based on global positioning system (GPS) tracked data.

Passage	No. of Sites	Total Area (ha)	Mean Area (ha)	Species	Usage in 2017 (days) *
Southern	32	462,254	14,445	All	142
Northeast	31	1,502,998	48,483	GWFG, LWFG, BG	649
Northwest	47	210,409	4476	GreyLag, SG, BG	191
Total	110	2,175,661	67,404	All	982

Notes. * accumulative days of stay in 2017. We only reported the usage in 2017 as it has the most routes (19 of 33). Moreover, we have all five geese in 2017. GWFG: Greater White-Fronted Goose, LWFG: Lesser White-Fronted Goose, BG: Bean Goose; SG: Swan Goose; Greylag: Greylag Goose.

**Table 2 ijerph-16-01147-t002:** Use types (%) within the stopover sites.

Passage	A	B	C	D	E	F
Southern	56.68	26.23	13.97	0.43	0.69	1.98
Northeast	72.00	14.83	9.05	2.32	0.48	1.32
Northwest	24.57	9.24	62.94	0.86	2.15	0.23
Migration path average	62.94	17.86	15.48	1.60	0.70	1.43

A = cropland, B = wetland, C = grassland, D = forest, E = bare-ground, and F = build-up area.

**Table 3 ijerph-16-01147-t003:** Summary of the protected stopover sites.

	No of Stopover Sites	Protected Area (ha)	Percentage (%)
I	1	3030	0.14
II	6	183,773	8.38
III	14	155,953	7.11
IV	89	1,849,641	84.37

I, satisfactory protected; II, under-protected; III, critically under-protected, and IV, unprotected.

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
