# Peer review of "Bird Satellite Tracking Revealed Critical Protection Gaps in East Asian–Australasian Flyway"

_ijerph, 2019, doi:10.3390/ijerph16071147_

Round 1

Reviewer 1 Report

EAAF is probably the most important flyway for waterbirds in China. Authors focus on the five geese to see if their flyways are covered enough by PAs, and try to link the conservation of waterbirds and development of agriculture. This is a good point, I have just a few comments here.

As we all know, birds migrate in spring and fall, so actually we need to address four seasons, wintering, spring migration, reproducing and fall migration, to cover the whole of waterbirds. Even only focus on stopover ecology, we need to include not only spring but also fall migration. I think the fall migration is at least equal to the spring, whether they fly the same way? Whether they use the same stopover sites? Conservation measures should be made on both spring and fall stopovers, not only one of them. SO if possible, I suggest adding the fall migration stopover datasets. Or at least give your reasons why only spring, and then the conservation and management suggestions shoud be modified accordingly.

In the abstract and introduction part, authors stated several times that stopover ecology in east asia is under-studied (line 16, 59-60). That might be true, but at least as I know, Prof. Ma Zhijun and Prof. Zhang Zhengwang have done a lot of works on the stopover ecology of shorebirds, Prof. Guo Yumin has done a lot on cranes, et al. or maybe authors just want to say stopover ecology of goose is few, then focus on your species. eg. line 26, I suggest replace 'migratory waterbirds' to 'geese'.

Line 185-186, there has published a paper last year by You et al. in PNAS (https://www.pnas.org/content/115/39/E9026), the big database of WDPA might lose a lot of key protected information of China, I am not sure if this occurs in your study, but please at least check. As I think, the study mainly in China, your group should have the database of nature reserves in China, that would be more comprehensive than WDPA. 

minor, line 113-115, repeated latin names of the five geese.

Author Response

Response to Reviewer 1 Comments

Point 1: Comments and Suggestions for Authors

EAAF is probably the most important flyway for waterbirds in China. Authors focus on the five geese to see if their flyways are covered enough by PAs, and try to link the conservation of waterbirds and development of agriculture. This is a good point, I have just a few comments here.

Response 1: We appreciate your professional reviewing work and the positive comments.

Point 2: As we all know, birds migrate in spring and fall, so actually we need to address four seasons, wintering, spring migration, reproducing and fall migration, to cover the whole of waterbirds. Even only focus on stopover ecology, we need to include not only spring but also fall migration. I think the fall migration is at least equal to the spring, whether they fly the same way? Whether they use the same stopover sites? Conservation measures should be made on both spring and fall stopovers, not only one of them. SO if possible, I suggest adding the fall migration stopover datasets. Or at least give your reasons why only spring, and then the conservation and management suggestions shoud be modified accordingly.

Response 2: Thanks for your comments. We totally agree that it would be ideal to add the autumn migration (paths and stopovers), however, the number of birds that have autumn migration data is rather small due to mortality or device failure, and we only have 20 birds return to China this year with data, which is insufficient for stopover analysis. Besides, we didn’t receive the GPS data when wrote the paper (due to device data roaming limitation, we can receive GSP locations only when the bird in in China – in method section). Furthermore, spring migration has direct effects on reproduction, thus is more important for the fitness of the species. Therefore we choose to focus on the spring migration first and will do more analysis as you suggested when we collected more data in the future. We added the reason above in the manuscript. (line 306-308)

Point 3: In the abstract and introduction part, authors stated several times that stopover ecology in east asia is under-studied (line 16, 59-60). That might be true, but at least as I know, Prof. Ma Zhijun and Prof. Zhang Zhengwang have done a lot of works on the stopover ecology of shorebirds, Prof. Guo Yumin has done a lot on cranes, et al. or maybe authors just want to say stopover ecology of goose is few, then focus on your species. eg. line 26, I suggest replace 'migratory waterbirds' to 'geese'.

Response 3: Thanks for your comments. Yes, there are a few studies on migratory shorebirds, and we acknowledged that in the Induction and Discussion. We changed migratory waterbirds to “migratory Anatidae” according to your suggestion. (line 64,66,69)

Point 4: Line 185-186, there has published a paper last year by You et al. in PNAS (https://www.pnas.org/content/115/39/E9026), the big database of WDPA might lose a lot of key protected information of China, I am not sure if this occurs in your study, but please at least check. As I think, the study mainly in China, your group should have the database of nature reserves in China, that would be more comprehensive than WDPA.

Response 4: Thanks for your comments. First, as we used all PA sites (not only classes IV and VI in You et al.), the discrepancy between WDPA and actual China’s protection network is small. Second, In You et al. (2018), there is a severe mistake in Table 1 – the total area of PAs in China, according to the Authors, is 1.47E+10 km2, which is more than 1,500 times of the total area of China (~ 9.6E+6 km2). In contrast, the percentage of total protected area in China is 14.6% according to WDPA, which is quite reasonable. Therefore, we believe that the WDPA database is more reliable. We also checked the WDPA database, and found that the missing sites are normally small urban parks, which are irrelevant with our study.

Point 5: minor, line 113-115, repeated latin names of the five geese.

Response 5: We are very sorry for our negligence and have deleted it.

Once again, thank you very much for your comments and suggestions.

Reviewer 2 Report

The authors cogently present important research regarding migratory habitat for at-risk geese species. I look forward to seeing the manuscript published. Aside from minor typographical errors, the writing and content are appropriate.

Author Response

Response to Reviewer 2 Comments

Point 1: The authors cogently present important research regarding migratory habitat for at-risk geese species. I look forward to seeing the manuscript published. Aside from minor typographical errors, the writing and content are appropriate.

Response 1: We appreciate your positive comments. We have checked the manuscript thoroughly, and corrected all typos.

Reviewer 3 Report

The ms describes an exploratory study of flight stopovers of 5 geese species between China and the Arctic, showing that most stopovers were in croplands.  The study appears to be very well executed and the results are both significant and interesting.  The ms is clearly written though with frequent minor grammatical errors.

My concern is that the ms is unnecessarily long and is not focused on the question being addressed.  Therefore I think each section except the abstract should be rewritten.

- Title is inappropriate.  Instead it's about type & protection level of geese stopovers

- Introduction: Far too general.  It should introduce (1) the species (first mentioned only in para 5; line 82), (2) importance of stopovers to conservation, and (3) land use patterns across the study area (currently not mentioned).

- Aim: last para of the introduction contains method, which should be moved.  3 aims are mentioned in the last sentence of the introduction indicating this is an exploratory study rather than an experiment.  Perhaps state a hypothesis ... that geese stopovers are mostly unprotected?

- Results & discussion: Put most important findings first, and remove unimportant ones.

- Conclusion: should just state the most important things (3 routes, most stopovers are in cropland ...)

Some specifics:

L32-34: This statement might be true but is debatable and should be avoided.  Maybe its due to better tools being available today.

L94: This is method, not introduction

L125: Better to say no *known* long-term habituation

L167: I think this should be demonstrated graphically

L186-189: Remove - no need to explain the database

L193-198: Combine into one line each as follows

    I. >75%   Totally / satisfactorily protected 

L255: Vertical axis not fully defined.  No reason to have abbreviations in horizontal axis 

L283: Results repeated in text & table

Author Response

Response to Reviewer 3 Comments

Point 1: The ms describes an exploratory study of flight stopovers of 5 geese species between China and the Arctic, showing that most stopovers were in croplands.  The study appears to be very well executed and the results are both significant and interesting.  The ms is clearly written though with frequent minor grammatical errors.

Response 1: Thanks for your supportive comments.

Point 2: My concern is that the ms is unnecessarily long and is not focused on the question being addressed.  Therefore I think each section except the abstract should be rewritten.

- Title is inappropriate.  Instead it's about type & protection level of geese stopovers

Response 2: Thanks for your comments. The title has been changed to “Bird tracking revealed critical protection gaps in East Asian-Australasian Flyway”

Point 3: - Introduction: Far too general.  It should introduce (1) the species (first mentioned only in para 5; line 82), (2) importance of stopovers to conservation, and (3) land use patterns across the study area (currently not mentioned).

Response 3: Thanks for your comments. We think the brief introduction on stopover ecology is necessary and relevant. The importance of including stopover sites in the conservation of migratory animals is often overlooked in current protection networks, especially in EAAF, and the main objective of the study is to raising awareness of whole life cycle protection.

According to your suggestion, we added the following paragraph about land cover:

“Wetlands in Eastern China forms part of the core areas of the EAAF. However, the rapid change of land cover, especially the expansion of urbanization, may pose a grave threat to the migratory species[5]. For example, as one of the main agricultural regions in China, 30% of Land in Northeast is arable, providing abundant food resource for migratory waterbirds[27]. However, the Northeast have experienced land cover change from 1990 to 2000, with farmland decreased by 386,195 ha and grassland decreased by 140,075ha[28]. With the loss of natural wetland and human induced land cover conversion, the protection of stopovers in Eastern China are facing great challenge.” (line 93-100)

Point 4: - Aim: last para of the introduction contains method, which should be moved.  3 aims are mentioned in the last sentence of the introduction indicating this is an exploratory study rather than an experiment.  Perhaps state a hypothesis ... that geese stopovers are mostly unprotected?

Response 4: Thanks. We deleted the method in introduction and added a hypothesis in the end of introduction:

“In this study, we focused on the spring migration as spring stopovers are critical for the population dynamics of Arctic geese[41], as they affect directly the reproduction success and survival of juveniles. Furthermore, we estimated the relative amount of land use along the migration route thus provides the foundation for conserving migratory routes and allows the quantitative assessment of current conservation efforts. Our hypothesis is that the spring stopovers of migration geese are mostly unprotected and the objectives of the study include: 1) determine the key stopover sites of five geese species in China; 2) the analysis of the main landscape type of those key stopover sites; and 3) evaluate protection gap and identify the critical shortfalls in Eastern Asian water bird conservation network.” (line 101-117)

Point 5:- Results & discussion: Put most important findings first, and remove unimportant ones.

Response 5: We arranged the results and discussion according to the objectives of the study.

Point 6:- Conclusion: should just state the most important things (3 routes, most stopovers are in cropland ...)

Response 6: Thanks. We revised the conclusion as below:

“In the study, we delineated three distinct migration passages: all the tagged geese shared the common southern passage to the wetlands and estuaries around Bohai Bay, from where short-distance migrants such as Swan Goose and Greylag Goose took the northwest routes to the temperate steppes; and the long-distance migrants (i.e. Lesser White Fronted Goose and Greater White Fronted Goose) flied northeast to the arctic tundra. Most of the stopover sites were in China, from where the geese travelled more or less directly to their breeding destinations. Stopovers sites were mainly covered by croplands and were severely under-protected by the current protection network. The revealed protection gaps may have contributed to the high proportion of threatened Anatidae species in the EAAF. In addition, our study could not establish the causality between agricultural land usage and goose habitat preference despite the tagged geese heavily used croplands.” (line 407-417)

Some specifics:

Point 7: L32-34: This statement might be true but is debatable and should be avoided.  Maybe its due to better tools being available today.

Response 7: We changed the sentences to

“The conservation of migratory birds has gained new momentum across the globe[1], mainly due to the recognition of their irreplaceable ecological functions and services[2], but also because of the widespread declines in their populations[3], and more sophistical tools to track their movement across the landscape [4]. (line 31-33)

Point 8: L94: This is method, not introduction

Response 8: We deleted the method and revised the paragraph. (line 101)

Point 9: L125: Better to say no *known* long-term habituation

Response 9: Thanks, changed to “has neglectable long-term habituation…” (line 138)

Point 10: L167: I think this should be demonstrated graphically

Response 10: Thanks for your suggestion. In the results, we provided such a graph showing the average days the geese spent in stopover site. (Figure2, left, line 278)

Point 11: L186-189: Remove - no need to explain the database

Response 11: Thanks for your comments, as WDPA is the key database we used to map the protection gaps, we think it might be better to briefly introduce it.

Point 12: L193-198: Combine into one line each as follows

I. >75%   Totally / satisfactorily protected

Response 12: Thanks. We revised as below: (line 206-210)

“We applied the following criteria to classify the stopover sites as below:

I.     >75%    Totally / satisfactorily protected;

II.   25-75%  Under-protected;

III.  < 25%   Critically under-protected; and

IV.   =0     unprotected.”

Point 13: L255: Vertical axis not fully defined.  No reason to have abbreviations in horizontal axis

Response 13: Thanks for your comments. We added the unit of day in the vertical axis and explained the figure more clearly in the captain “Summary graphs of the average days the geese spent in stopover site (left) and the home range the geese used in stopover site (right).” Because the full name of the geese species are too long to present in the X-axis, so we used the abbreviations and explained it in the captain.

Point 14: L283: Results repeated in text & table

Response 14: We deleted the repeat part.

Once again, thank you very much for your comments and suggestions.

Round 2

Reviewer 1 Report

No more comments.

Author Response

Response to Reviewer 2 Comments

Point 1: No more comments.

Response 1: We feel great thanks for your professional review work on our manuscript.

Reviewer 3 Report

Manuscript was already good, and has been improved with this revision.

I started making notes, but there are further grammatical changes needed after line 45.

23 - Area*s*

24 – sites *are*

24 – fulfil (not fulfill)

25 – enabling

33 – sophisticated

39 – avoid competition, escape predation, breed, and to take advantage of spatiotemporal resource variation

41 – requires a

41 – encountered

45 – first mention of all species should cite authority, e.g.; Limosa lapponica (Linnaeus, 1758), thereafter L. lapponica.  I use this website: http://www.catalogueoflife.org/col/

44-45 – sentence needs rewriting

81 – I think it’d be better to remove species common names and acronyms from the ms and just use scientific names  … including Anser erythropus (Linnaeus, 1758), Anser albifrons (Scopoli, 1769), etc.  Note that genus should be provided at each first mention of a species.

And then ..

124 – “including A. erythropus (Lesser White-fronted Goose), A. albifrons (Greater White-fronted Goose)

Author Response

Response to Reviewer 3 Comments

Point 1: Manuscript was already good, and has been improved with this revision.

Response 1: We appreciate your professional review work on our manuscript.

Point 2: I started making notes, but there are further grammatical changes needed after line 45.

23 - Area*s*

Response 2: corrected accordingly (line 23).

Point 3: 24 – sites *are*

Response 3: corrected accordingly (line 24).

Point 4: 24 – fulfil (not fulfill)

Response 4: corrected accordingly. (line 24)

Point 5: 25 – enabling

Response 5: corrected accordingly. (line 26)

Point 6: 33 – sophisticated

Response 6: corrected accordingly. (line 33)

Point 7: 39 – avoid competition, escape predation, breed, and to take advantage of spatiotemporal resource variation

Response 7: Thanks, we revised the sentence accordingly. (line40-41)

Point 8: 41 – requires a

Response 8: corrected accordingly. (line 42)

Point 9: 41 – encountered

Response 9: corrected accordingly. (line 43)

Point 10: 45 – first mention of all species should cite authority, e.g.; Limosa lapponica (Linnaeus, 1758), thereafter L. lapponica.  I use this website: http://www.catalogueoflife.org/col/

Response 10: Thanks for the detailed comment, we added the authority for all species the first time we mentioned. (line 48,85-87,359,388-389)

Point 11: 44-45 – sentence needs rewriting

Response 11: Thanks, we revised the sentence as “Conservation actions targeting part of life circle (e.g. breeding, nonbreeding or stopover sites) may not be able to prevent population decline of these birds.” (line 43-46)

Point 12: 81 – I think it’d be better to remove species common names and acronyms from the ms and just use scientific names … including Anser erythropus (Linnaeus, 1758), Anser albifrons (Scopoli, 1769), etc.  Note that genus should be provided at each first mention of a species.

Response 12: Thanks for your suggestion. We think it’s acceptable and widely practiced to using common name if the scientific name is given at the first time the species is mentioned. We used acronyms to save space.

Point 13: And then ..

124 – “including A. erythropus (Lesser White-fronted Goose), A. albifrons (Greater White-fronted Goose)

Response 13: Thanks for your suggestion. Please see our response to Point 13.

Once again, thank you very much for your comments and suggestions.
